# SpaceTGN: Augmented Mini-Batch Negative Sampling for Continuous-Time Dynamic Graph Learning

## Abstract

Continuous-Time Dynamic Graph (CTDG) learning has significantly advanced link prediction performance by leveraging random negative sampling and incorporating adaptive temporal information. Recent studies aim to improve performance by introducing random sampling to obtain hard negative samples, whose quality is limited by randomness, capturing few categories of negative samples, and leading to false positive (FP) and false negative (FN) problems. Here we present SpaceTGN, a CTDG learning framework, with a augmented hard negative sampling mini-batches (AMNS) strategy and two new feature extraction strategies that derive space-temporal locality subgraph and historical occurrence information to emphasize the graph's temporal discriminative properties. The AMNS strategy sample mini-batches comprised of instances that are hard-to-distinguish (i.e., hard and true negatives with respect to each other) based on the target distribution, thereby effectively augmenting the discriminative features and the diversity of historical and inductive samples. Furthermore, to mitigate the challenges posed by false positives and false negatives, our architecture SpaceTGN employs a conceptually straightforward approach that investigates temporal subgraphs and historical interactions between source and destination nodes. This enables the model to leverage complex and historically accurate interactions among predicted entities. Our extensive evaluation of dynamic link prediction on seven state-of-the-practice datasets reveals that SpaceTGN achieves state-of-the-art performance in most datasets, demonstrating its effectiveness in ameliorating model bias.

## 1 Introduction

Dynamic graph modeling offers a versatile representation of real-world scenarios by abstracting entities as nodes and the time-varied interactions or relationships between these entities as temporal edges. This modeling framework is applicable to a wide range of domains, including social networks Kumar et al. (2019); Huo et al. (2018); Alvarez-Rodriguez et al. (2021), traffic networks Zhao et al. (2019); Yu et al. (2017); Wu et al. (2019); Guo et al. (2019); Yu et al. (2021), the recommendation system Song et al. (2019); Dong et al. (2012); Wang et al. (2021b), and financial transactions Wang et al. (2021c); Zhang et al. (2022); Feng et al. (2019). To facilitate efficient learning on dynamic graphs, many efforts have been devoted to the development of discrete-time dynamic graph models (DTDG) Zhao et al. (2019); Pareja et al. (2020); Yang et al. (2021) and continuous-time dynamic graph models (CTDG) Kumar et al. (2019); Rossi et al. (2020); Cong et al. (2022); Yu et al. (2024).

Despite the state-of-the-art (SOTA) work achieving continuous optimization and nearly perfecting many existing benchmark datasets, they exhibit two notable limitations: *only select negative samples at a "random" level* and *difficult to capture periodic dependency and historical occurrence informations*. Firstly, most of them Yu et al. (2024); Kumar et al. (2019); da Xu et al. (2020); Cong et al. (2022); Wang et al. (2021a) widely exploit random negative sampling to improve the efficiency and effectiveness of CTDG learning, but this strategy often results in overfitting of the model. Their negative sampling strategy randomly selects destination nodes from the entire set of nodes, retaining the timestamps, features, and source nodes of positive edges. However, this strategy introduces extreme variation between the negative and positive edges, leading to overfitting problems in recent models. The models, after training, can only judge the datasets with obvious positive and negative

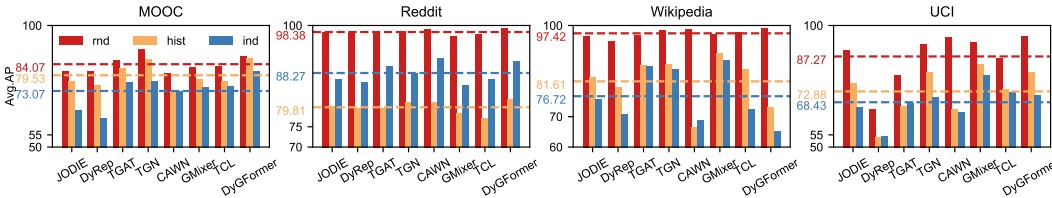

Figure 1: Average AP for transductive dynamic link prediction in recent work with random(rnd), historical(hist), and inductive(ind) measurements. GMixer is the abbreviation for GraphMixer.

differences. Poursafaei et al. (2022) presented the first investigation to introduce negative sampling measurements (historical and inductive negative sampling) to robust dynamic graph link prediction, revealing the suboptimal performance of recent models and the weak generality. For example, Figure 1 illustrates that the performance of most SOTA methods decreases significantly (more than 11%) when different negative samplings (random, historical and inductive) are introduced in the test time. State-of-the-art models frequently tend to overfit positive examples while neglecting negative ones, greatly limiting their applicability in real-world scenarios. Therefore, we conclude that *it is necessary for selecting real hard negtive samples to balance the divergence in negative sample distribution in the training and testing stages*.

Secondly, existing studies Cong et al. (2022); Yu et al. (2024); Tian et al. (2024) exploit direct neighbor sampling to capture the characteristics of the graph structure, access first-hop neighbors, and employ random walks to generate wandering sequences without exploiting periodic subgraph patterns. The lack of historical and inductive sampling information makes it difficult to distinguish hard negative samples and leads to information discrimination. For example, existing studies mainly capture the historical neighbors of node $u$ and $v$ separately without modeling their periodic indirect interactions. When extracting periodic timing information, current methods only capture local neighbor interactions, resulting in loss of global information on the graph such as message passing da Xu et al. (2020), memory networks Kumar et al. (2019); Trivedi et al. (2019); Rossi et al. (2020), and feature encoding Yu et al. (2024); Tian et al. (2024). As a result, previous methods cannot still effectively capture *the periodic dependency and historical occurrence information.*

In this paper, we design a new CTDG learning framework, namely SPACETGN. Internally, we outline two following key technical contributions in SPACETGN to address the above challenges:

**We propose a new augmented mini-batch negative sampling (AMNS) strategy for better continuous-time graph learning.** The central idea of this paper is to sample mini-batches of hard-to-distinguish instances to emphasize CTDG's discriminative temporal properties. Specifically, we generate these mini-batches to cover hard yet true negatives by dynamically maintaining a collection of previously encountered, high-smilarity temporal edges and persistently sampling from this collection. This hard negative sampling strategy is integrated with a sample augmentation module that uses a targeted distribution to enrich the data set, which improves diversity among negative samples for training.

**We present a novel Time-Sequence-based dynamic graph learning model (SPACETGN).** Two CTDG features are explicitly utilized to capture periodic and historical interactions in SPACETGN: *space-temporal dependency* and *historical occurrence*. The space-temporal locality dependency captures the interaction sequence of the current node preceding a specific time point, which represents the temporal local features. This sequence is then combined with the historical neighbor sequence of the node, reflecting the local spatial features. Furthermore, we introduce a historical occurrence encoding scheme to capture the factual interactions among predicted entities. SPACETGN introduces an optimized MLP-Mixer layer to distill the intrinsic features of the extracted sequences, which significantly improves the capture of temporal information in the model.

From our empirical validation, SPACETGN significantly outperforms existing SOTA methods on most datasets, proving the efficacy of our design and coding strategies. Furthermore, all models are evaluated for their statistical performance that is significantly higher than previous results in both historical and inductive scenarios, confirming the effectiveness of our negative sampling strategy.

## 2 RELATED WORK

**Continuous-Time Dynamic Graph Learning Architectures.** Dynamic graph neural networks can be broadly classified into discrete-time dynamic graphs (DTDG) Zhao et al. (2019); Pareja et al.

(2020); Sankar et al. (2020); You et al. (2022) and continuous-time dynamic graphs (CTDG) Kumar et al. (2019); Rossi et al. (2020); Trivedi et al. (2019); da Xu et al. (2020); Wang et al. (2020). CTDG approaches Yu et al. (2024); Tian et al. (2024); Rossi et al. (2020) depict dynamic graphs as chronologically ordered interaction lists, offering a more flexible, general, and challenging paradigm for representation learning. In the context of CTDG, models typically capture neighbor sequences of nodes utilizing foundational frameworks such as Recurrent Neural Networks or Self-Attention mechanisms, exemplified by TGAT da Xu et al. (2020) and JODIE Kumar et al. (2019). Furthermore, storage update-based models include TGN Rossi et al. (2020) and APAN Wang et al. (2021c); CAWN Wang et al. (2020) utilizes a random walk strategy; models leveraging ordinary differential equations Liang et al. (2022) and temporal point processes Chang et al. (2020); GraphMixer Cong et al. (2022) employs a purely MLP-based architecture; DyGFormer Yu et al. (2024) integrates a Transformer-based Vaswani et al. (2017) approach; and FreeDyG Tian et al. (2024) incorporates Fourier transform techniques.

**Negative Sampling on CTDG Learning.** Negative sampling uses a selected subset of non-observed or negative data points to significantly improve training efficiency and model performance, which is widely used in various domains, including natural language processing Grbovic et al. (2015); Zhang & Zweigenbaum (2018); Wu et al. (2021), computer vision Wu et al. (2017); Robinson et al. (2020); Wu et al. (2020), and recommendation systems Rendle et al. (2009; 2012). The existing work has been classified into two lines: *static negative sampling* and *hard negative sampling*. The initial approach, static negative sampling, involves assigning a fixed probability to each dynamic candidate for selection. This includes methods such as random negative sampling (RNS) and popularity-based negative sampling. In contrast, hard negative sampling emphasizes the selection of challenging negative samples, which are characterized by their similarity to positive samples within dynamic distributions.Yang et al. (2020); Shrivastava et al. (2016), as a specialized variant of negative sampling, offer more informative training signals, allowing for a more precise characterization of dynamic characteristics. Inspired by this technique, we utilized hard negative samples to mitigate significant discrepancies observed in the model performance across random, historical, and inductive continuous-time link prediction scenarios. This method not only addresses the inherent variability in model testing conditions, but also improves the robustness and accuracy of our learning algorithms under diverse operational settings.

## 3 PRELIMINARIES

**Definition 3.1 (Dynamic Graph)** *We define the set of nodes as $N$ and the collection of temporal edges as $E = \{(u_1, v_1, t_1), (u_2, v_2, t_2), \ldots\}$, where the timestamps follow an ascending order, $0 \leq t_1 \leq t_2 \leq \ldots$. Here, $u_i$ and $v_i$ from $N$ represent the nodes originating and terminating at the edge $i^{th}$ at the time instant $t_i$, respectively. Each node $u \in N$ is endowed with a node feature $x_u \in \mathbb{R}^{d_N}$, whereas each edge $e \in E$ carries an edge feature $x_e \in \mathbb{R}^{d_E}$. Together, the set of nodes and edge features are denoted by $F_N$ and $F_E$, respectively. Note that if the raw dynamic graph inputs lack node and edge features, these features are default configured to a null vector. To encapsulate, we conceptualize the continuous-time dynamic graph as $G = \{N, E, F_N, F_E\}$, formalizing temporal interactions within graph-structured data.*

**Definition 3.2 (Continuous-Time Link Prediction)** *Given a dynamic graph $G$ and a source node $u \in N$, a destination node $v \in N$, a specific timestamp $t$, along with a set of historical edges $E = \{(u', v', t') | t' < t\}$ that precede $t$, the objective of edge prediction is to design algorithms capable of deriving representations $h_u^t \in \mathbb{R}^d$ and $h_v^t \in \mathbb{R}^d$ for nodes $u$ and $v$, respectively, where $d$ denotes the dimensionality of these representations. The ultimate goal is to use these representations to infer the likelihood of an edge between $u$ and $v$ at the timestamp $t$.*

## 4 PROPOSED METHOD

This section details our framework SPACETGN that incorporates two key designs: an augmented mini-batch negative sampling strategy and a new architecture for CTDG learning. In particular, the architecture is newly designed with two novel feature extraction for space-temporal locality dependency and historical occurrence.

## 4.1 AUGMENTED MINI-BATCH NEGATIVE SAMPLING STRATEGY

Previous methods Yu et al. (2024); Tian et al. (2024); Rossi et al. (2020) randomly selected destination nodes from the entire set of nodes for negative sampling, as illustrated in Figure 2. This approach often results in negative samples that do not have a connection to positive samples, making them easy to distinguish. However, in real-world applications, it is essential to consider negative samples that have relationships similar to the positive samples to challenge the model's discriminative capabilities.

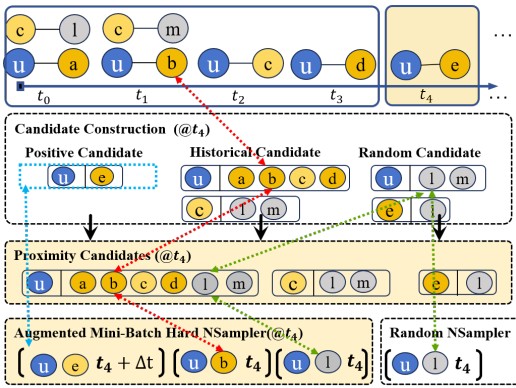

Figure 2: A Comparative Study of Augmented Mini-Batch Hard Negative Sampler (with highlight color) and Random Negative Sampler (without highlight color). The process encompasses two stages: the construction of proximity candidates and the sampling of instances.

**Paradigm.** To achieve robust CTDG learning, we propose the augmented mini-batch negative sampling (AMNS) paradigm, initially optimizing the alignment of the distributions of positive and negative samples. The truly hard negative samples are selected to exhibit a high likelihood of interaction yet are absent in the observed data. Figure 2 gives an illustration of the augmented mini-batch negative sampling on graph $G$. AMNS executed based on mini-batches satisfies the following three key criteria:

- **Criteria 1: The hardness of a selected negative sample should have a high predicted intensity.** The hard negative sample should be negatively correlated with the frequency of interaction of the historical-aware pair of nodes. In contrast to setting predetermined random negative samples for each training epoch as described in Huang et al. (2024), the AMNS approach is anticipated to ascertain higher priority levels of hardness based on the frequency of historical occurrences.

- **Criteria 2: The hardness of a selected negative sample should be negatively related to its positive sample of high similarity.** First, since historical events encompass extended timestamps, it is advisable to select negative samples characterized by lower degrees of hardness Poursafaei et al. (2022). In contrast, as the temporal proximity to the prediction instance increases, the selection of negative samples with higher degrees of hardness becomes pertinent, which can expedite the optimization of positive samples while ensuring that negative samples of greater difficulty remain temporally recent, thereby mitigating the false-positive challenge.

- **Criteria 3: The proportional distribution between the positive samples and the selected hard negative samples should be adaptive adjustive.** To cover the diverse variety of real-world CTDG learning, for example, different datasets or evaluation metrics , the addition of random samples can improve the robustness of the model. Therefore, the ratio of HNS (hard negative samples) to SNS (soft negative samples), as well as the ratio of positive samples to hard negative samples in each minibatch, can be adaptively adjusted.

**Instantiation of Augmented Mini-Batch Negative Sampling** We give a concrete instantiation of our new hard negative sampling strategy (HNS), which involves incorporating selected adaptive mini-batch negative samples (as shown in Figure 2). We mark three sets of candidates: the positive candidate as $E_{\text{pos}}^t$, the historical candidate as $E_{\text{hist}}^t$, and the random candidate as $E_{\text{rnd}}^t$. Our AMNS dynamically constructs the proximity candidate set $E_{\text{pro}}^t = E_{\text{hist}}^t \cup E_{\text{rnd}}^t \setminus E_{\text{pos}}^t$ . Then, during the augmented mini-batch hard negative sampling stage, the edges absent at the current moment are identified as negative examples by sampling the set $E_{\text{pro}}^t$ , $E_{\text{rnd}} - E_{\text{pos}}^t$ and $E_{\text{pro}}^t$ following the formula 1. Taking the dynamic graph sample in Figure 2 as an example, the positive sample $(u, b, t_{11}) \in E^{t_{11}}$, the previous negative sample $(u, h, t_{11}) \in E_{\text{rnd}}$, and the hard negative sample $(u, c, t_{11}) \in E_{\text{pro}}^{t_{11}}$.

Consequently, we propose three metrics to evaluate the specified criteria and the sample $E_{\text{pro}}^t$: 1) we measure the intensity term based on recency: $\phi_{\text{int}}(u, v, t) = \exp\left(-\delta \cdot (t - t_{\text{last}}^{(u,v)})\right)$; 2) the

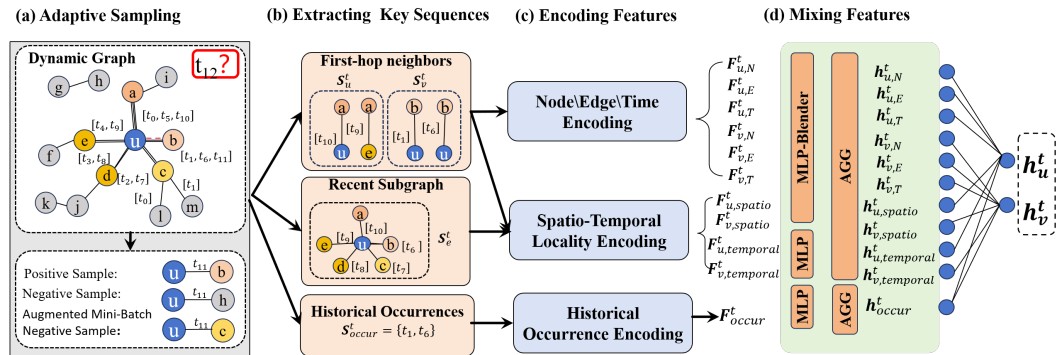

Figure 3: Framework of the proposed , consisting of four modules: (a) Adaptive Sampling, (b) Extracting Key Sequences, (c) Encoding Features, and (d) Mixing Features.

similarity ranking of the pair of nodes $(u, v)$ is measured as $\phi_r(u, v, t) = r_{(u,v)}(t)$, indicating their relative likelihood of interaction compared to other pairs; 3) given the node embeddings $\mathbf{h}_u(t)$ and $\mathbf{h}_v(t)$, the similarity measure can be measured as $\phi_{sim}(u, v, t) = \mathbf{h}_u(t)^\top \mathbf{h}_v(t)$. In addition, we design an innovative optimal proportional distribution function within the AMNS framework. For each candidate negative sample, as discussed in $E_{pro}^t - E_t$, the rating function is defined as follows:

$$\lambda_{(u,v)}^*(t) = exp\left(\alpha \cdot \exp\left(-\delta \cdot (t - t_{last}^{(u,v)})\right) + \beta \cdot \left(1 - \frac{r_{(u,v)} - 1}{N - 1}\right) + \gamma \cdot \mathbf{h}_u(t)^\top \mathbf{h}_v(t) + b\right) \tag{1}$$

where $\alpha > 0$, $\beta > 0$, $\gamma > 0$ are predefined hyperparameters to represent the proportion of $E_{hist}^t$, $E_{rnd}^t$, and $E_{pos}^t$, $t_{last}^{(u,v)}$ is the time of the last interaction between $u$ and $v$, and $\delta$ is a decay rate, $r_{(u,v)}$ as the relative likelihood of interaction between $u$ and $v$. After calculating and filtering the high scores of all proximity structures, we obtain a set of ratings $HNS$, and then the final set of negative samples is identified. This proximity makes hard negative samples particularly valuable for training, as they push the model to learn more nuanced distinctions between similar pairs, enhancing its overall discriminatory power and effectiveness.

## 4.2 SPACE-TEMPORAL LOCALITY AWARE CTDG ARCHITECTURE

Compared with the existing CTDG negative sampling strategy, our proposed negative sampling is closer to the positive samples, which poses a greater challenge to the model's information extraction capabilities. Existing models mostly employ message passing mechanisms to capture topological features or to aggregate information from low-order neighboring nodes; nevertheless, deriving temporal locality and periodicity remains particularly challenging. We propose two types of information extraction methods based on current feature extraction: **Space-Temporal Locality** and **Historical Occurrence**, which significantly emphasize the temporal discriminative properties of CTDG.

SPACETGN, as illustrated in Figure 3, begins by extracting three key sequences for a given source node $u$, destination node $v$, and timestamp $t$: the historical sequences of the first hop neighbors for both nodes, the sequences of dynamic edges occurring before $t$, and the temporal sequences marking the occurrence of the edge $(u, v)$ in historical data.

- **Extracting the historical sequences of the first hop neighbors for both nodes $u$ and $v$.** Given the predicted nodes $u$, $v$, and timestamp $t$, we extract the historical sequences of the first hop neighbors for $u$ and $v$, denoted as $S_u^t$ and $S_v^t$, respectively, where $S_u^t = \{(u, u', t')|t' < t\} \cup \{(u', u, t')|t' < t\}$ and $S_v^t = \{(v, v', t')|t' < t\} \cup \{(v', v, t')|t' < t\}$ Yu et al. (2024). We introduce a parameter $l$ to represent the maximum length of first-hop neighbor sequences. If a sequence exceeds $l$, excess neighbors are truncated. If a sequence is shorter than $l$, it is padded with zero vectors to reach the required length.

- **Extracting the sequences of dynamic edges occurring before the timestamp $t_i$.** Existing methods primarily capture the historical neighbors of node $u$ and $v$ with-

out modeling their periodic indirect interactions. In contrast, we present a temporal dependency extraction to capture the periodic interaction subgraph in a fixed time window, denoted $S_e^t$. The subgraph is composed of edge sequences $S_e^t = \{(u_{i-r}, v_{i-r}, t_{i-r}), (u_{i-r+1}, v_{i-r+1}, t_{i-r+1}), \ldots, (u_{i-1}, v_{i-1}, t_{i-1})\}$, where the time window series are ordered such that $t_{i-r} \leq t_{i-r+1} \leq \cdots \leq t_{i-1} < t_i$. Here, $r$ is introduced as a tunable parameter that governs the size of the window, configured by the breadth of the local subgraph.

- **Extracting the temporal sequences marking the occurrences of the edge** $(u, v)$ **in historical data.** To handle edge prediction in dynamic graphs, previous methodologies, both discrete-time and continuous-time models, usually overlooked the significance of a node's historical occurrences for feature extraction. However, temporal features reveal crucial cyclic patterns and information interaction dynamics. To bridge this gap, we introduce the extraction of historical occurrence time series, denoted as $S_{occur}^t = \{t_i | (u_i, v_i, t_i), u_i = u, v_i = v, t_i < t\}$, for each pair of nodes individually. Similarly, we introduce a parameter $o$ to indicate the maximum sequence length. Sequences longer than $o$ are truncated, while shorter sequences are padded with zero vectors to the desired length.

After encoding the extracted sequences, SPACETGN learn intricate correlations within the features through an MLP-Mixer layer. The MLP-Mixer output representations are then aggregated via a fully connected sequence aggregation layer, integrating various feature representations into a unified representation that captures the temporal dynamics of the nodes at time $t$. Using learnable parameters, the model combines these features to construct a holistic temporal perception for the nodes $u$ and $v$. Finally, a Link Prediction layer executes a probabilistic forecast to determine the likelihood of a connection between nodes $u$ and $v$ at the given timestamp.Detailed descriptions of SPACETGN are available in Section A.

## 5 EXPERIMENTS

In this section, we present experimental results that demonstrate that our proposed negative sampling method surpasses the random negative sampling methods utilized in DyGLib and TGB Huang et al. (2024). Detailed descriptions of these baselines are available in Section D. In addition, we provide a comprehensive analysis of negative sampling techniques, focusing on their impact on the distribution of sampled data, as well as their implications for model training.Finally, by implementing our improved negative sampling method, we conducted a comprehensive comparison between our proposed SPACETGN and the eight state-of-the-art models in DyGLib in the seven publicly available real-world datasets. Our experiments demonstrated that SPACETGN consistently outperforms existing models in extracting valuable information, highlighting the advantages of utilizing dynamic graphs for information extraction.

### 5.1 EXPERIMENTAL SETTINGS

To evaluate our model's performance, we adhere to the established benchmarks in the field by utilizing Average Precision (AP) and Area Under the Receiver Operating Characteristic Curve (AUC-ROC) as primary metrics. The experimental setup encompasses two distinct scenarios for link prediction: (1) *transductive setting*, whose goal is to predict the formation of an edge between two nodes, both of which have been observed during the training phase, and (2) *inductive setting*, which aims to predict edge formation involving at least one node that was not present during the training phase. We note that a node is considered inductive if it does not appear in the training data. To facilitate training, validation, and testing, we split these datasets into three chronological segments with ratios of 70%/15%/15%.

We optimized all Adam Kingma & Ba (2014) models (excluding EdgeBank, which has no trainable parameters). We train the models 100 times over time and use an early stopping strategy with a patience value of 20. We select the model with the best performance on the validation set for testing. We configure all methods' learning rate and batch size on all datasets to 0.0001 and 200, respectively. We run these methods with 0 to 4 seeds five times and report the average performance to eliminate bias. The parameters of the recent models are detailed in the Section F. Experiments were performed on an NVIDIA GeForce RTX4080 16GB GPU device.

## 5.2 Effectiveness of Negative Sampling Approach

We validate the efficacy of the negative sampling enhanced training method across various datasets from DyGLib by incorporating AMNS with three distinct models: TCLWang et al. (2021a), Graph-MixerCong et al. (2022), and DyGFormerYu et al. (2024). We quantify the performance of methods in terms of average precision (AP) in historical measurements by evaluating their implementation w/ AMNS and w/o AMNS, in which Table 1 reports their performance. Table 5 in Section E.1 shows further their performance in terms of average precision (AP) under inductive measurements.

We find that TCL, GraphMixer, and DyGFormer usually produce substantial improvements in various methods and datasets after integrating our advanced negative sampling strategy, achieving an average improvement of 16.18%, 11.85%, and 9.44%. The TCL method exhibits a marked performance improvement, benefiting from the AMNS approach. In particular, TCL w / AMNS can achieve an improvement of 46.26%, with AP soaring from 59.30 to 86.73 on the LastFM dataset, which shows the best negative sampling enhancement. These significant advances verify the effectiveness of our self-adaptive hard negative sampling approach and highlight the importance of capturing discriminative properties of positive and negative samples.

Table 1: AP in hist for different methods when equipped with AMNS. Note that the AP results are multiplied by 100 for a better display layout.

| Datasets | TCL | | | GraphMixer | | | DyGFormer | | |
|---|---|---|---|---|---|---|---|---|---|
| | Original | Enhanced | Improv. | Original | Enhanced | Improv. | Original | Enhanced | Improv. |
| Wikipedia | 85.78 | 92.55 | 7.89% | 90.76 | 94.14 | 3.72% | 73.10 | 94.50 | 22.65% |
| Reddit | 77.18 | 82.43 | 6.80% | 78.25 | 89.44 | 14.30% | 81.71 | 90.54 | 9.75% |
| MOOC | 77.08 | 96.15 | 24.74% | 78.01 | 96.91 | 24.23% | 86.53 | 97.94 | 11.65% |
| LastFM | 59.30 | 86.73 | 46.26% | 72.47 | 91.82 | 26.70% | 81.57 | 88.79 | 8.13% |
| Enron | 72.79 | 78.88 | 8.37% | 78.07 | 84.42 | 8.13% | 76.86 | 79.55 | 3.38% |
| Social Evo. | 95.96 | 99.31 | 3.49% | 95.00 | 99.25 | 4.47% | 97.09 | 99.49 | 2.41% |
| UCI | 73.91 | 85.50 | 15.68% | 83.98 | 85.68 | 2.02% | 80.91 | 87.92 | 7.97% |

## 5.3 Impact of Negative Sampling on Model Training

We conducted AMNS analysis experiments on the wiki dataset using TGB. DyGFormer achieved state-of-the-art mean reciprocal rank (MRR) with the original random sampling method. Upon applying AMNS to DyGFormer, we observed improvements of over 1% in both Validation MRR and Test MRR, as shown in the Table 2.

Fig 4 illustrates the comparison of the MRR metrics of DyGFormer under the original random negative sampling (RND) and the proposed negative sampling method (AMNS) during the training phase. The experimental results show that the validation MRR of the RND method reaches a peak (about 0.82) in the first 10 epochs, and then there is almost no enhancement with large fluctuations, which shows its limitation for model learning. In contrast, the AMNS method continues to increase after 20 epochs and achieves a new peak ( 0.84) at 40 epochs, and the validation MRR then remains stable. This suggests that AMNS effectively facilitates the model's learning of difficult samples, significantly improves performance, and demonstrates its advantages and potential for improvement during training. Thus, AMNS is able to better guide model learning and enhance its generalization ability compared to the traditional RND negative sampling method.

Table 2: Comparison of MRR Metrics for DyGFormer: Original Random Negative Sampling (RND) vs. Proposed Negative Sampling (AMNS) on the tgbl-wiki Dataset

| Method | Validation MRR | Test MRR |
|---|---|---|
| RND | 81.60 ± 0.50 | 79.80 ± 0.40 |
| AMNS | 85.36 ± 0.25 | 81.12 ± 0.19 |

## 5.4 Impact of Negative Sampling on Sample Distribution

In this section, we analyze the distribution of positive and negative samples from a single training run using the wiki dataset from TGB. We utilize the historical occurrences of positive and negative samples as statistics to present the distribution plots for positive samples, original random negative

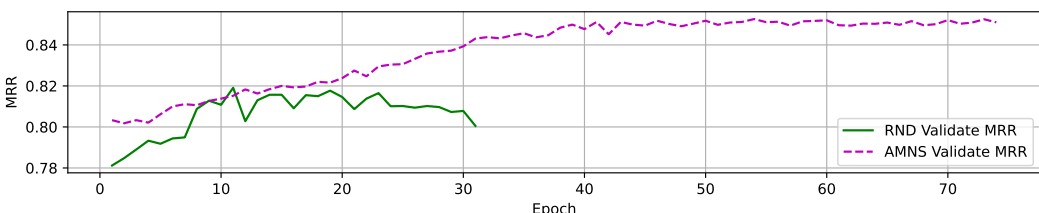

Figure 4: Comparison of MRR Metrics for DyGFormer during Traning Stage: Original Random Negative Sampling (RND) vs. Proposed Negative Sampling (AMNS) on the tgbl-wiki Dataset

sampling, and AMNS negative sampling. As shown in Figure 6, the Log Distribution of Positive

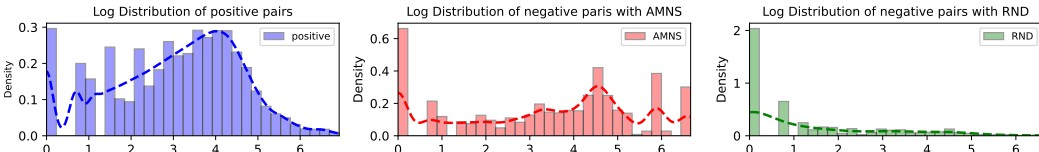

Figure 5: Distribution of Positive Samples, Original Random Negative Sampling, and AMNS Negative Sampling Based on Historical Occurrences in the Wiki Dataset from TGB

Pairs shows a pronounced peak around log values of 1 to 4, indicating a substantial concentration of occurrences in this range.In comparison, the Log Distribution of Negative Pairs with AMNS exhibits a distribution that closely resembles that of the positive pairs, particularly in the log value range of 1 to 5. This proximity highlights AMNS's ability to capture hard negative samples, which challenge the model effectively and facilitate stronger learning.Conversely, the Log Distribution of Negative Pairs with RND demonstrates a stark contrast, with a rapid decline in density as log values increase. Most of the negative samples are concentrated at lower log values, suggesting that they provide fewer informative challenges for the model during training.

To further illustrate that AMNS captures negative samples with a higher degree of similarity to positive samples compared to RND, we conducted an analysis comparing the cosine similarity of historical occurrences for both AMNS and RND across each epoch, as shown in the figure 6. The

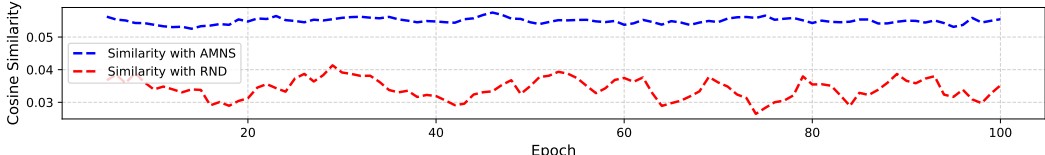

Figure 6: Comparison of Cosine Similarity of Historical Occurrences Between Positive Samples and Negative Samples Captured by AMNS and RND Across Epochs

results reveal that the cosine similarity for AMNS remains consistently higher and more stable, peaking around 0.056, indicating that AMNS effectively captures hard negative samples that closely align with the positive samples. In contrast, the similarity values for RND fluctuate significantly between 0.030 and 0.040, suggesting that the random sampling approach yields less relevant negative samples that may not contribute effectively to model learning. This analysis underscores the superiority of AMNS in enhancing the model's performance by selectively sampling informative negatives, thereby facilitating improved discernment of relationships during training.

## 5.5 PERFORMANCE COMPARISON WITH BASELINES AND DISCUSSIONS

We report the performance of recent methods by comparing our SPACETGN framework against the baseline methods in both transductive and inductive dynamic link prediction. Table 3 presents the average precision scores (APs) metric for all datasets under random, historical, and inductive measurements, as proposed in Poursafaei et al. (2022).

First, we observe that SPACETGN usually achieves higher accuracy than the baselines, with an average rank ranking of *1.29/1.0/1.14* across the three measurements, respectively. And, for the UCI

Table 3: AP for transductive link prediction under random, historical, and inductive measurements. We denote the best and second-best results by emphasizing **bold** and underlined fonts. Note that the results are multiplied by 100 for a better display layout. MS is the abbreviation of measurements.

| MS | Datasets | JODIE | DyRep | TGAT | TGN | EdgeBank | TCL | GraphMixer | DyGFormer | SPACETGN |
|---|---|---|---|---|---|---|---|---|---|---|
| rnd | Wikipedia | 93.54 ± 0.49 | 91.54 ± 0.30 | 95.69 ± 0.20 | 96.71 ± 0.06 | 90.37 ± 0.00 | 95.26 ± 0.19 | 95.99 ± 0.04 | 98.06 ± 0.10 | **98.41 ± 0.10** |
| | Reddit | 96.36 ± 0.38 | 96.72 ± 0.20 | 97.54 ± 0.03 | 97.4 ± 0.13 | 94.86 ± 0.00 | 95.96 ± 0.06 | 95.36 ± 0.08 | **98.10 ± 0.05** | 98.08 ± 0.12 |
| | MOOC | 81.34 ± 1.52 | 81.17 ± 0.57 | 85.75 ± 0.09 | **90.39 ± 1.27** | 57.97 ± 0.00 | 83.22 ± 1.28 | 82.86 ± 0.18 | 87.55 ± 0.30 | 89.23 ± 0.08 |
| | LastFM | 70.42 ± 1.08 | 71.34 ± 1.77 | 71.35 ± 0.11 | 76.16 ± 2.63 | 79.29 ± 0.00 | 75.81 ± 0.64 | 73.78 ± 0.18 | 90.77 ± 0.14 | **90.95 ± 0.20** |
| | Enron | 82.00 ± 3.18 | 79.29 ± 2.71 | 68.29 ± 1.75 | 85.81 ± 1.54 | 83.53 ± 0.00 | 85.65 ± 0.45 | 80.82 ± 0.42 | 91.33 ± 0.33 | **91.78 ± 0.28** |
| | Social Evo | 89.17 ± 0.77 | 88.96 ± 0.13 | 92.74 ± 0.09 | 92.93 ± 0.36 | 74.95 ± 0.00 | 93.64 ± 0.20 | 92.59 ± 0.09 | 94.48 ± 0.03 | **94.66 ± 0.03** |
| | UCI | 84.73 ± 1.38 | 50.80 ± 5.25 | 76.42 ± 1.75 | 87.24 ± 0.86 | 76.20 ± 0.00 | 85.81 ± 5.87 | 88.44 ± 2.01 | 93.90 ± 0.23 | **94.16 ± 0.38** |
| | Avg. Rank | 7.00 | 7.57 | 5.71 | 3.29 | 7.43 | 5.00 | 5.71 | 2.00 | **1.29** |
| hist | Wikipedia | 88.92 ± 0.62 | 85.25 ± 0.45 | 91.63 ± 0.19 | 92.57 ± 0.45 | 73.09 ± 0.00 | 92.55 ± 0.46 | 94.14 ± 0.80 | 94.50 ± 0.47 | **96.92 ± 0.12** |
| | Reddit | 88.18 ± 0.39 | 83.46 ± 0.67 | 84.02 ± 0.13 | 87.26 ± 0.31 | 73.66 ± 0.00 | 82.43 ± 0.14 | 89.44 ± 0.07 | 90.54 ± 0.13 | **93.03 ± 0.25** |
| | MOOC | 94.03 ± 0.54 | 84.69 ± 3.42 | 94.43 ± 0.49 | 97.42 ± 0.41 | 60.71 ± 0.00 | 96.15 ± 0.26 | 96.91 ± 0.13 | 97.94 ± 0.20 | **98.80 ± 0.19** |
| | LastFM | 82.07 ± 1.35 | 76.29 ± 2.96 | 79.60 ± 0.97 | 77.43 ± 3.88 | 73.21 ± 0.00 | 86.73 ± 1.28 | 91.82 ± 0.09 | 88.79 ± 0.35 | **94.60 ± 0.25** |
| | Enron | 80.32 ± 2.09 | 74.13 ± 2.55 | 67.16 ± 2.04 | 74.52 ± 1.12 | 76.90 ± 0.00 | 78.88 ± 0.48 | 84.42 ± 0.71 | 79.55 ± 0.76 | **89.59 ± 0.37** |
| | Social Evo | 89.56 ± 3.35 | 95.23 ± 0.16 | 99.03 ± 0.06 | 98.79 ± 0.37 | 80.60 ± 0.00 | 99.31 ± 0.06 | 99.25 ± 0.05 | 99.49 ± 0.01 | **99.74 ± 0.01** |
| | UCI | 90.53 ± 0.07 | 49.70 ± 5.38 | 81.05 ± 1.81 | 85.92 ± 0.83 | 65.03 ± 0.00 | 85.50 ± 6.16 | 85.68 ± 2.81 | 87.92 ± 1.24 | **97.67 ± 0.26** |
| | Avg. Rank | 5.14 | 7.86 | 6.43 | 5.14 | 8.43 | 5.14 | 3.29 | 2.57 | **1.00** |
| ind | Wikipedia | 83.75 ± 0.34 | 82.41 ± 0.86 | 91.19 ± 0.46 | 92.76 ± 0.48 | 80.65 ± 0.00 | 91.37 ± 0.30 | 91.15 ± 1.26 | 94.77 ± 0.47 | **95.05 ± 0.27** |
| | Reddit | 86.48 ± 0.91 | 84.16 ± 1.37 | 90.74 ± 0.12 | 87.96 ± 0.72 | 85.57 ± 0.00 | 88.48 ± 0.11 | 88.33 ± 0.13 | **92.86 ± 0.43** | 92.25 ± 0.22 |
| | MOOC | 79.82 ± 0.73 | 66.16 ± 3.67 | 89.61 ± 0.57 | 92.46 ± 1.14 | 49.44 ± 0.00 | 92.71 ± 0.35 | 91.81 ± 0.29 | 92.61 ± 0.37 | **93.68 ± 0.52** |
| | LastFM | 70.31 ± 2.31 | 65.58 ± 2.27 | 78.74 ± 0.96 | 71.18 ± 5.15 | 75.47 ± 0.00 | 76.30 ± 1.30 | 85.14 ± 0.15 | 83.31 ± 0.70 | **85.16 ± 0.65** |
| | Enron | 74.86 ± 3.66 | 70.44 ± 2.00 | 65.19 ± 2.36 | 73.02 ± 2.91 | 73.91 ± 0.00 | 76.11 ± 0.53 | 79.44 ± 0.69 | 80.38 ± 0.53 | **86.37 ± 0.23** |
| | Social Evo | 90.66 ± 2.87 | 95.29 ± 0.17 | 98.97 ± 0.06 | 98.92 ± 0.31 | 83.70 ± 0.00 | 99.28 ± 0.07 | 99.16 ± 0.06 | 99.52 ± 0.01 | **99.74 ± 0.01** |
| | UCI | 71.13 ± 0.17 | 53.56 ± 1.35 | 78.21 ± 1.10 | 71.87 ± 1.92 | 57.41 ± 0.00 | 81.74 ± 4.36 | 81.33 ± 1.43 | 78.83 ± 1.79 | **92.35 ± 0.35** |
| | Avg. Rank | 7.00 | 8.29 | 5.29 | 5.57 | 7.86 | 3.43 | 4.00 | 2.43 | **1.14** |

dataset with inductive measurement, SPACETGN achieves an accuracy of 92.35%, which is much higher (*10.6%*) than the second-ranked 81.74%. The reasons behind this phenomenon are that (i) the self-adaptive negative sampling approach (AMNS) and pattern extraction help SPACETGN extract more discriminative information from the original dynamics and negative samples, and (ii) the two temporal locality dependency and historical occurrence extraction strategies allow SPACETGN fully capture the distinct temporal feature differences between positive and negative samples. Second, in Table 3, our evaluations show that SPACETGN achieves significant performance improvement across historical and inductive measurements than other baselines. This is because the AMNS in SPACETGN improves the model's discriminative ability during the training process.

We also present the results of the Average Precision (AP) for inductive dynamic link prediction and the AUC-ROC (Area Under the Receiver Operating Characteristic Curve) for both transductive and inductive predictions in Section E.2 and E.3. From these results, we find SPACETGN usually achieves better performance, with average rankings around $1 - 1.57$, especially rank $1 - 1.14$ across historical and inductive ones. Therefore, we conclude that SPACETGN consistently obtains better performance than most baselines, achieving an impressive average rank of 1.2 among them, further demonstrating its superiority and the effectiveness of AMNS approach.

Further, we conduct ablation studies on SPACETGN to validate the efficacy of our proposed temporal locality and historical occurrence feature extraction. The detailed results of these experiments can be found in Section E.4. Our model, SPACETGN, achieves the highest performance (93.34%-96.75%) with the two feature extraction methods. Performance declines without this feature extraction. In conclusion, our Space-Temporal feature extraction effectively distills accurate temporal information, while the Historical Occurrence feature extraction captures the dynamic graph's cyclical information. Together, these tailored methods demonstrate both necessity and effectiveness.

## 6 CONCLUSION

This paper introduces a new self-adaptive negative sampling approach to effectively mitigate the overfitting issues previously encountered in continuous-time dynamic graph learning. Leveraging the negative sampling approach, we developed SPACETGN, an MLP-Mixer architecture incorporating comprehensive encoding strategies. Our model has demonstrated state-of-the-art (SOTA) performance across seven publicly available datasets under three distinct negative sampling strategies. Our work offers a fresh perspective by addressing the dynamic graph edge prediction problem through the lens of dynamic feature mining and feature fusion.

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

# 7 APPENDIX

# A DETAILS OF SPACETGN.

## A.1 ENCODING FEATURES

**Space-Temporal Dependency Encoding.** Inspired by the neighbor co-occurrence encoding Yu et al. (2024), we propose an advanced encoding strategy named *Space-Temporal Locality Dependency Encoding*. This approach captures the topological structure and integrates the temporal dynamics better for discriminating hard negative samples (see Section E.4).

*Space Locality Dependency.* For space locality encoding, we quantify the frequency of neighbors within historical sequences $S_u^t$ and $S_v^t$, encoding this data into frequency features Yu et al. (2024). E.g., the neighbors of nodes $u$ and $v$ are $\{a, a, b, b, c\}$ and $\{a, b, c, d, e\}$ respectively, with interaction frequencies $\{a : 2, b : 2, c : 1, d : 0, e : 0\}$ for $u$ and $\{a : 1, b : 1, c : 1, d : 1, e : 1\}$ for $v$, the dependency sequences are encoded as $F_u^t = [[2, 1], [2, 1], [1, 1], [0, 1], [0, 1]]^T$ and $F_v^t = [[1, 1], [1, 1], [1, 1], [1, 1], [1, 1]]^T$. These sequences are further processed through a function $f(\cdot)$, which maps inputs into a $d_{\text{space}}$ dimensional space using a two-layer perceptron with ReLU activation:

$$F_{*,\text{space}}^t = f(F_*^t[:, 0]) + f(F_*^t[:, 1]) \in \mathbb{R}^{L \times d_{\text{space}}}, \tag{2}$$

where $*$ can represent either nodes $u$ or $v$.

*Temporal Locality Dependency.* The temporal locality encoding calculates the amount of interactions in $S_u^t$ and $S_v^t$ relative to $S_e^t$ detailed as sequences $S_{u,e}^t$ and $S_{v,e}^t$ for source and destination nodes respectively. These counts are transformed using the function $f(\cdot)$ into a $d_{\text{temporal}}$ dimensional space, generating outputs $F_{u,\text{temporal}}^t \in \mathbb{R}^{L \times d_{\text{temporal}}}$ for node $u$, and $F_{v,\text{temporal}}^t \in \mathbb{R}^{L \times d_{\text{temporal}}}$ for node $v$.

**Historical Occurrence Encoding.** Upon extracting the historical occurrences time series $S_{occur}^t$, we apply time encoding to generate $F_{occur}^t \in \mathbb{R}^{o \times d_{occur}}$, where $d_{occur}$ represents the dimension dedicated to capturing the temporal patterns of occurrences. This encoding scheme is particularly beneficial when applied to enhanced hard negative sample learning, as it provides a robust framework for distinguishing between genuine and artificially introduced patterns in the data. This method benefits a better understanding of the underlying temporal dynamics and significantly improves the model's ability to generalize from both positive and negative examples in the dataset.

## A.2 MIXING FEATURES

**Single-Channel Mixing.** Upon finalizing the feature encoding process, each feature is independently processed through the MLP-Mixer, which is structured to elucidate the inherent relationships within the sequences. The MLP-Mixer is articulated through two principal equations:

$$F_{token} = F_{token} + W_{token}^2 \cdot \text{ReLU}\left(W_{token}^1 \cdot \text{LayerNorm}(input)\right), \tag{3}$$

$$F_{channel} = F_{token} + \text{ReLU}\left(\text{LayerNorm}(F_{token}) \cdot W_{channel}^1\right) \cdot W_{channel}^2, \tag{4}$$

where the input pertains to $F_{u,*}^t \in \mathbb{R}^{l \times d}$ and $F_{v,*}^t \in \mathbb{R}^{l \times d}$, spanning feature categories such as node ($N$), edge ($E$), time ($T$), topological structure ($space$), temporal dynamics ($temporal$), and occurrence ($F_{occur}^t \in \mathbb{R}^{o \times d}$). Here, $W_{token}^*$ and $W_{channel}^*$ denote the adjustable parameters in the token-mixing and channel-mixing MLPs, respectively. $F_{channel}$ is the output of the MLP-Mixer.

**Multi-Channel Aggregation.** We employ a fully connected layer with learnable parameters for aggregation. Specifically, $W^{\text{agg},*} \in \mathbb{R}^{ld \times d_{output}}$ and $b^{\text{agg}} \in \mathbb{R}^{d_{output}}$ are utilized to aggregate features, resulting in:

$$h_{u,*}^t = F_{u,*}^t \cdot W_{u,*}^{\text{agg}} + b^{\text{agg}}, \tag{5}$$

$$h_{v,*}^t = F_{v,*}^t \cdot W_{v,*}^{\text{agg}} + b^{\text{agg}}, \tag{6}$$

where $*$ signifies the feature types ($N$, $E$, $T$, $space$, $temporal$). For the occurrence features $F_{occur}^t$, aggregation is performed similarly, employing $W^{\text{agg},*} \in \mathbb{R}^{od \times d_{output}}$, resulting in:

$$h_{\text{occur}}^t = F_{\text{occur}}^t \cdot W_{\text{occur}}^{\text{agg}} + b^{\text{agg}}. \tag{7}$$

### A.3 Loss Function

For edge prediction tasks, we employ the cross-entropy loss function to measure the discrepancy between the true labels and the predicted probabilities. The loss function is defined as follows:

$$\text{Loss} = -\frac{1}{K} \sum_{i=1}^{S} \left( y_i \log(\hat{y}_i) + (1 - y_i) \log(1 - \hat{y}_i) \right), \tag{8}$$

where $K$ denotes the number of positive and negative samples, $y_i$ represents the true label, and $\hat{y}_i$ signifies the predicted probability for each sample.samples, optimizing the model's performance in predicting the existence of edges within the dynamic graph.

## B  Complexity analysis of SpaceTGN

In our analysis, we adopt a batch-wise approach to evaluate the computational complexity of SpaceTGN, where $b$ denotes the batch size and $N$ represents the number of temporal edges involved.

**Extracting Key Sequences.**Initially, each node within a batch retrieves its first-hop neighbors, entailing a bisection lookup with a complexity of $O(\log N)$. Subsequently, the selection of neighboring nodes incurs a complexity of $O(l)$, where $l$ is the length of the neighbors list. Thus, the total complexity for processing one batch in this phase is $O(b \times (\log N + l))$. For the extraction of the Space-Temporal Series and Historical Occurrences sequences, the complexities are computed as $O(b \times (\log N + br))$ and $O(b \times (\log N + bo))$, respectively, where $r$ and $o$ represent the lengths of the respective sequences.

**Encoding Features.**The complexities for Node/Edge encoding are $O(b \times l \times d_N)$ and $O(b \times l \times d_E)$, where $d_N$ and $d_E$ are the original dimensions of each feature, respectively. Time Encoding incurs a complexity of $O(b \times l \times d_T)$, where $d_T$ is the dimension of the time feature. Space-Temporal Dependency Encoding is characterized by $O(b \times (l \log l + r \log r + l + r)) + O(l \times d_{\text{temporal}} + l \times d_{\text{temporal}}^2)$ and $O(b \times (l \log l + l)) + O(l \times d_{\text{space}} + l \times d_{\text{space}}^2)$, where $d_{\text{temporal}}$ and $d_{\text{space}}$ are the dimensions of temporal locality and space locality features, respectively. Historical Occurrence Encoding features a complexity of $O(b \times o \times d_{\text{occur}})$, where $d_{\text{occur}}$ represents the dimension dedicated to capturing the temporal patterns of occurrence.

**Mixing Features.**The MLP-Mixer processes these features with complexities $O(b \times (l^2 + d^2))$ and $O(b \times (o^2 + d^2))$. Finally, the aggregation layer, which merges features for the final output, incurs complexities of $O(l \times d \times d_{\text{output}})$ and $O(o \times d \times d_{\text{output}})$, where $d_{\text{output}}$ represents the output dimension.

## C  DETAILS OF DATASETS.

Table 4: Dataset statistics

| Dataset | Nodes | Edges | Unique Edges | Node/Link Feature | Time Granularity | Duration |
|---------|-------|-------|--------------|-------------------|------------------|----------|
| Wikipedia | 9227 | 157474 | 18257 | 0/172 | Unix timestamp | 1 month |
| Reddit | 10984 | 672447 | 78516 | 0/172 | Unix timestamp | 1 month |
| MOOC | 7144 | 411749 | 178443 | 0/4 | Unix timestamp | 17 months |
| LastFM | 1980 | 1293103 | 154993 | 0/0 | Unix timestamp | 1 month |
| Enron | 184 | 125235 | 3125 | 0/0 | Unix timestamp | 3 years |
| Social Evo. | 74 | 2099519 | 4486 | 0/2 | Unix timestamp | 8 months |
| UCI | 1899 | 59835 | 20296 | 0/0 | Unix timestamp | 196 days |

We use seven datasets collected by Poursafaei et al. (2022) in the experiments, which are publicly available:

**Wikipedia**: This dataset comprises edits made to Wikipedia pages over the course of one month, modeling both editors and Wiki pages as nodes, with edges represented by timestamped edit requests. Edge features include LIWC feature vectors derived from the text of the edits.

**Reddit**: This dataset captures the interactions within a one-month period on Reddit, where nodes represent users or posts, and edges denote timestamped posting actions.

**MOOC**: This graph represents interactions between students and online course content units, such as problem sets and videos. Each edge in the graph signifies a student's access to a specific content unit, mapping the educational engagement over a certain period.

**LastFM**: In this dataset, users and songs are modeled as nodes, with edges capturing the relationship of users listening to songs. It encompasses the interactions of 1000 users with the 1000 most listened to songs within a one-month timeframe. The graph is non-attributed, focusing solely on user-song interactions.

**Enron**: This dataset is comprised of approximately 50,000 emails exchanged among employees of the Enron Corporation over a span of three years, forming a complex network of email correspondence.

**Social Evo.**: This mobile phone proximity graph tracks the daily interactions within an entire undergraduate dormitory from October 2008 to May 2009, reflecting the social dynamics and connectivity patterns.

**UCI**: This dataset represents an on-line communication network similar to Facebook among students at the University of California, Irvine, with edges timestamped to the second, offering a granular view of social interactions over time.

These datasets provide a rich basis for analyzing dynamic networks on different temporal scales and in varying contexts, which is ideal for comprehensive studies of network dynamics and behavior modeling in social, educational, and corporate settings.

# D    DETAIL DESCRIPTIONS OF BASELINES.

In this study, we evaluate the performance of our models against eight baseline methods, each uniquely designed to handle dynamic graph data:

**JODIE**: This model is tailored for temporal bipartite networks, specifically user-item interactions. Using two synchronized recurrent neural networks, it continuously updates the state of the user and the item. In addition, it integrates a projection operation to model the future trajectory of the representation of each entity Kumar et al. (2019).

**DyRep**: This approach features a recurrent architecture that updates node states after each interaction. It incorporates a temporal-attentive aggregation module to account for the evolving structural information within dynamic graphsTrivedi et al. (2019).

**TGAT**: The Temporal Graph Attention Network leverages the self-attention mechanism to aggregate features from each node's temporal-topological neighbors. It includes a time-encoding function to capture temporal patterns effectivelyda Xu et al. (2020).

**TGN**: This model maintains a dynamic memory for each node, updating it upon observing new interactions via a combination of message function, aggregator, and memory updater. A dedicated embedding module generates temporal representations for the nodesRossi et al. (2020).

**EdgeBank**: This is a memory-based, parameter-free method for the prediction of transductive dynamic links. It manages observed interactions within a memory unit and updates this memory via several strategies. The system categorizes an interaction as positive if retained in memory, and negative otherwisePoursafaei et al. (2022).

**TCL**: The Temporal Contextual Linking model begins by generating each node's interaction sequence through a breadth-first search on the temporal dependency interaction subgraph. It then employs a graph transformer that integrates graph topology and temporal information, enhancing its learning capabilities with a cross-attention mechanism for the interdependencies between interacting nodes Wang et al. (2021a).

**GraphMixer**: This model demonstrates the efficacy of a fixed time encoding function over a trainable version. It integrates this function into a link encoder based on the MLP-Mixer architecture to

analyze temporal links, while a node encoder with neighbor mean-pooling summarizes node featuresCong et al. (2022).

**DyGFormer** introduces a novel architecture based on the Transformer framework. DyGFormer primarily leverages the first-hop interactions of nodes through two innovative techniques: a neighbor co-occurrence encoding scheme, which explores the correlations between source and destination nodes based on their historical sequences, and a patching technique, which divides each sequence into multiple patches before feeding them into the Transformer, thereby enabling the model to benefit effectively and efficiently from longer historiesYu et al. (2024).

These baselines provide a comprehensive set of tools for benchmarking dynamic graph analysis methods, offering insights into the various strategies for handling temporal and structural changes in network data.

# E    DETAILED EXPERIMENTAL RESULTS

## E.1    ADDITIONAL RESULTS FOR NEGATIVE SAMPLING

In Table 5, we present the performance of each model (TCL, GraphMixer, DyGFormer) in terms of average precision (AP) under inductive settings. We compare the results both before (original model) and after (enhanced model) the implementation of our training method. This comparison illustrates the impact of our hard negative sampling strategy on improving model performance.

Table 5: AP in inductive for different methods when equipped with the negative sampling augmentation training method.Note that the results AP are multiplied by 100 for a better display layout.

| Datasets | TCL | | | GraphMixer | | | DyGFormer | | |
|---|---|---|---|---|---|---|---|---|---|
| | Original | Enhanced | Improv. | Original | Enhanced | Improv. | Original | Enhanced | Improv. |
| Wikipedia | 72.53 | 91.37 | 25.98% | 88.56 | 91.15 | 2.92% | 65.27 | 94.77 | 45.20% |
| Reddit | 86.80 | 88.48 | 1.94% | 85.26 | 88.33 | 3.60% | 91.29 | 92.86 | 1.72% |
| MOOC | 74.87 | 92.71 | 23.83% | 74.66 | 91.81 | 22.97% | 81.17 | 92.61 | 14.09% |
| LastFM | 58.21 | 76.30 | 31.08% | 68.12 | 85.14 | 24.99% | 73.56 | 83.31 | 13.25% |
| Enron | 71.69 | 76.11 | 6.17% | 74.63 | 79.44 | 6.45% | 78.18 | 80.38 | 2.81% |
| Social Evo | 96.12 | 99.28 | 3.29% | 94.85 | 99.16 | 4.54% | 97.52 | 99.52 | 2.05% |
| UCI | 72.63 | 81.74 | 12.54% | 79.62 | 81.33 | 2.15% | 71.46 | 78.83 | 10.31% |

Table 5 illustrates the significant impact of our advanced negative sampling augmentation training method on average precision (AP) scores across various models and datasets. The TCL method achieves an overall improvement of 14.97%, with the LastFM dataset showing a remarkable increase of 31.08%, raising the AP from 58.21 to 85.14. GraphMixer also benefits, displaying an average improvement of 9.66%, particularly excelling on LastFM with a 24.99% enhancement. DyGFormer records a 12.78% average improvement, with the Wikipedia dataset achieving the highest increase of 45.20%. These results validate the efficacy of our AMNS approach and emphasize its role in enhancing the discriminative properties of positive and negative samples.

## E.2    ADDITIONAL RESULTS FOR TRANSDUCTIVE DYNAMIC LINK PREDICTION

We show the AUC-ROC for transductive dynamic link prediction under random, historical, and inductive measurements in Table 6. Our proposed model, SPACETGN, consistently outperforms the baselines, with an average rank of *1.57/1.0/1.29*. In particular, it achieves an accuracy of 90.50% on the UCI dataset under inductive measurement, exceeding the second-ranked model by 12.69% (77.81%). The outstanding performance of SPACETGN validates (i) the efficacy of the self-adaptive negative sampling approach (AMNS) and pattern extraction techniques in capturing more discriminative information, and (ii) the effectiveness of temporal locality dependency and historical occurrence strategies in distinguishing positive and negative samples.

## E.3    ADDITIONAL RESULTS FOR INDUCTIVE DYNAMIC LINK PREDICTION

We present the AP and AUC-ROC for inductive dynamic link prediction with three negative sampling strategies in Table 7 and Table 8. Our model, SPACETGN, consistently performs well relative to the baselines, achieving an average rank of *1.29/1.14/1.14* across the measurements. Specifically,

Table 6: AUC-ROC for transductive link prediction under random, historical, and inductive measurements. The best and second-best results are emphasized by bold and underlined fonts. Note that the results are multiplied by 100 for a better display layout. MS is the abbreviation of measurements.

| MS | Datasets | JODIE | DyRep | TGAT | TGN | EdgeBank | TCL | GraphMixer | DyGFormer | SPACETGN |
|---|---|---|---|---|---|---|---|---|---|---|
| rnd | Wikipedia | 93.13 ± 0.46 | 90.99 ± 0.36 | 95.27 ± 0.22 | 96.50 ± 0.08 | 90.78 ± 0.00 | 94.36 ± 0.25 | 95.56 ± 0.08 | 97.90 ± 0.07 | **98.28 ± 0.12** |
| | Reddit | 96.50 ± 0.25 | 96.78 ± 0.15 | 97.55 ± 0.03 | 97.53 ± 0.11 | 95.37 ± 0.00 | 96.10 ± 0.05 | 95.42 ± 0.07 | **98.11 ± 0.05** | 97.99 ± 0.14 |
| | MOOC | 80.06 ± 2.07 | 82.16 ± 1.84 | 85.90 ± 0.20 | **92.23 ± 0.78** | 60.86 ± 0.00 | 82.71 ± 0.13 | 82.27 ± 0.11 | 85.47 ± 0.53 | 87.69 ± 0.28 |
| | LastFM | 69.01 ± 0.82 | 70.44 ± 1.65 | 68.78 ± 0.13 | 77.02 ± 2.27 | 83.77 ± 0.00 | 68.94 ± 0.82 | 70.48 ± 0.28 | **90.90 ± 0.12** | 90.58 ± 0.21 |
| | Enron | 85.03 ± 2.54 | 81.62 ± 2.45 | 66.30 ± 1.94 | 87.42 ± 1.18 | 87.05 ± 0.00 | 83.51 ± 0.81 | 83.01 ± 0.44 | 91.81 ± 0.37 | **92.47 ± 0.39** |
| | Social Evo | 91.40 ± 0.52 | 90.72 ± 0.11 | 94.23 ± 0.10 | 94.64 ± 0.42 | 81.60 ± 0.00 | 95.17 ± 0.12 | 94.53 ± 0.08 | 95.97 ± 0.03 | **96.37 ± 0.01** |
| | UCI | 87.43 ± 0.55 | 52.65 ± 7.39 | 74.96 ± 2.88 | 87.37 ± 0.84 | 77.30 ± 0.00 | 83.99 ± 5.64 | 86.07 ± 2.24 | **92.19 ± 0.34** | 92.14 ± 0.58 |
| | Avg. Rank | 6.14 | 7.29 | 6.14 | 3.29 | 7.14 | 5.86 | 5.71 | 1.86 | **1.57** |
| hist | Wikipedia | 87.09 ± 0.58 | 83.81 ± 0.48 | 88.93 ± 0.28 | 90.38 ± 0.50 | 77.10 ± 0.00 | 89.78 ± 0.70 | 92.14 ± 1.23 | 93.37 ± 0.58 | **96.32 ± 0.16** |
| | Reddit | 86.71 ± 0.33 | 82.26 ± 0.33 | 82.43 ± 0.15 | 85.45 ± 0.33 | 78.63 ± 0.00 | 80.64 ± 0.17 | 87.82 ± 0.09 | 89.43 ± 0.18 | **92.43 ± 0.29** |
| | MOOC | 94.19 ± 0.57 | 87.41 ± 2.61 | 93.83 ± 0.55 | 97.36 ± 0.43 | 61.90 ± 0.00 | 95.24 ± 0.28 | 96.71 ± 0.15 | 97.72 ± 0.26 | **98.79 ± 0.19** |
| | LastFM | 83.28 ± 1.15 | 76.24 ± 3.23 | 74.62 ± 1.06 | 77.53 ± 3.17 | 78.22 ± 0.00 | 83.24 ± 2.38 | 92.04 ± 0.13 | 87.34 ± 0.34 | **94.22 ± 0.32** |
| | Enron | 83.79 ± 2.25 | 77.18 ± 2.90 | 63.93 ± 2.79 | 76.46 ± 0.79 | 79.83 ± 0.00 | 76.87 ± 1.41 | 85.67 ± 0.77 | 79.33 ± 0.82 | **87.29 ± 0.60** |
| | Social Evo | 94.13 ± 1.04 | 95.20 ± 0.19 | 98.86 ± 0.06 | 98.62 ± 0.45 | 85.83 ± 0.00 | 99.20 ± 0.08 | 99.20 ± 0.06 | 99.39 ± 0.02 | **99.73 ± 0.01** |
| | UCI | 90.59 ± 0.06 | 49.35 ± 7.71 | 74.73 ± 3.16 | 85.01 ± 0.86 | 69.13 ± 0.00 | 81.46 ± 7.39 | 80.34 ± 3.65 | 85.66 ± 1.16 | **97.31 ± 0.30** |
| | Avg. Rank | 4.86 | 7.57 | 7.00 | 5.29 | 7.71 | 5.43 | 3.43 | 2.71 | **1.00** |
| ind | Wikipedia | 79.54 ± 0.56 | 79.00 ± 0.83 | 88.08 ± 0.60 | 90.24 ± 0.62 | 81.74 ± 0.00 | 88.07 ± 0.46 | 87.53 ± 2.05 | 93.06 ± 0.68 | **93.61 ± 0.37** |
| | Reddit | 83.29 ± 0.80 | 81.14 ± 0.69 | 88.00 ± 0.14 | 84.88 ± 0.59 | 85.97 ± 0.00 | 85.72 ± 0.13 | 85.88 ± 0.13 | **91.00 ± 0.48** | 90.96 ± 0.24 |
| | MOOC | 78.60 ± 0.96 | 67.30 ± 3.58 | 88.09 ± 0.80 | 91.95 ± 1.05 | 48.17 ± 0.00 | 92.11 ± 0.37 | 91.29 ± 0.33 | 92.02 ± 0.40 | **93.75 ± 0.53** |
| | LastFM | 69.53 ± 2.03 | 63.40 ± 2.33 | 73.75 ± 1.06 | 69.31 ± 4.12 | 77.36 ± 0.00 | 73.64 ± 2.36 | **84.78 ± 0.15** | 80.77 ± 0.38 | 81.35 ± 1.01 |
| | Enron | 76.49 ± 3.58 | 71.56 ± 2.80 | 61.25 ± 2.77 | 72.75 ± 2.79 | 75.03 ± 0.00 | 73.71 ± 1.16 | 79.01 ± 0.87 | 76.76 ± 0.64 | **82.68 ± 0.54** |
| | Social Evo | 94.56 ± 0.81 | 95.19 ± 0.20 | 98.80 ± 0.06 | 98.77 ± 0.37 | 87.88 ± 0.00 | 99.17 ± 0.08 | 99.12 ± 0.07 | 99.43 ± 0.02 | **99.72 ± 0.01** |
| | UCI | 69.84 ± 0.08 | 51.62 ± 1.04 | 72.14 ± 1.87 | 66.36 ± 2.12 | 57.99 ± 0.00 | 77.55 ± 5.33 | 77.81 ± 1.52 | 74.92 ± 1.74 | **90.50 ± 0.56** |
| | Avg. Rank | 6.86 | 8.43 | 5.29 | 6.00 | 6.57 | 4.43 | 3.57 | 2.57 | **1.29** |

under the UCI dataset with inductive measurement, SPACETGN records an AP of 93.04% and a AUC-ROC of 91.91%, significantly higher than the second-best score.

The strong performance of SPACETGN can be attributed to (i) the self-adaptive negative sampling approach (AMNS) and pattern extraction techniques, which enhance the extraction of discriminative information, and (ii) effective utilization of temporal locality dependency and historical occurrence strategies in differentiating positive and negative samples.

Moreover, the results in Table 7 highlight that SPACETGN exhibits substantial improvements across historical and inductive measurements compared to other methods. This effectiveness is primarily due to AMNS, which strengthens the model's discriminative ability during training, thereby solidifying its efficacy in link prediction tasks.

### E.4 EFFECTIVENESS OF TEMPORAL-SPACE LOCALITY AND HISTORICAL OCCURRENCE FEATURE EXTRACTION

We conduct ablation studies on SPACETGN to validate the efficacy of our proposed temporal locality and historical occurrence feature extraction. Specifically, we assess the impact of the feature extraction by comparing the standard SPACETGN model against variants where one or both of these optimizations are not incorporated. The variants are labeled as follows: without Space-Temporal Locality (w/o ST), without Historical Occurrence Encoding (w/o HO), and without both types of coding (w/o Both).

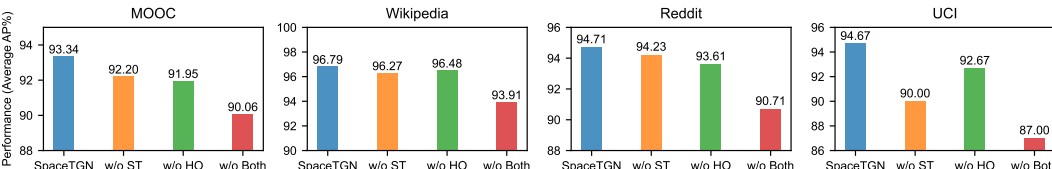

Figure 7: Ablation study of SPACETGN, where w/o ST and w/o HO represent SPACETGN without space-temporal locality dependency encoding and historical occurrence encoding techniques respectively. The performance is an average AP score under three measurements(rnd, hist, ind).

The detailed results of these experiments are illustrated in Figure 7. We observe that SPACETGN obtains the best performance (93.34% − 96.75%) when using the two feature extraction, and the re-

Table 7: AP for inductive link prediction under random, historical, and inductive measurements. The best and second-best results are emphasized by bold and underlined fonts. Note that the results are multiplied by 100 for a better display layout. MS is the abbreviation of measurements.

| MS | Datasets | JODIE | DyRep | TGAT | TGN | TCL | GraphMixer | DyGFormer | SPACETGN |
|---|---|---|---|---|---|---|---|---|---|
| rnd | Wikipedia | 91.26 ± 0.62 | 88.39 ± 0.52 | 94.87 ± 0.14 | 95.68 ± 0.18 | 95.02 ± 0.23 | 95.31 ± 0.21 | 97.29 ± 0.08 | **97.85 ± 0.12** |
| | Reddit | 92.26 ± 0.93 | 93.41 ± 0.83 | 95.27 ± 0.12 | 95.57 ± 0.27 | 91.40 ± 0.10 | 92.63 ± 0.17 | 96.90 ± 0.11 | **97.30 ± 0.16** |
| | MOOC | 77.51 ± 0.65 | 78.97 ± 1.23 | 84.32 ± 0.30 | **90.19 ± 0.95** | 81.60 ± 0.26 | 79.63 ± 0.16 | 84.30 ± 0.53 | 86.94 ± 0.13 |
| | LastFM | 82.21 ± 1.62 | 83.60 ± 1.37 | 76.71 ± 0.23 | 80.35 ± 1.43 | 80.36 ± 0.62 | 80.38 ± 0.25 | 92.18 ± 0.19 | **92.88 ± 0.17** |
| | Enron | 78.41 ± 1.69 | 74.29 ± 1.61 | 65.07 ± 1.39 | 77.98 ± 2.76 | 82.94 ± 0.41 | 74.84 ± 0.32 | 87.72 ± 0.63 | **87.76 ± 0.33** |
| | Social Evo | 91.72 ± 0.53 | 89.82 ± 0.75 | 90.93 ± 0.06 | 89.79 ± 0.37 | 92.22 ± 0.17 | 90.87 ± 0.14 | **92.97 ± 0.08** | 92.91 ± 0.04 |
| | UCI | 69.51 ± 2.00 | 48.02 ± 3.33 | 76.60 ± 1.03 | 82.73 ± 0.61 | 83.65 ± 6.02 | 87.30 ± 1.48 | 92.21 ± 0.21 | **92.98 ± 0.34** |
| | Avg. Rank | 5.86 | 6.43 | 5.71 | 4.57 | 4.86 | 5.14 | 2.14 | **1.29** |
| hist | Wikipedia | 78.34 ± 0.34 | 73.56 ± 0.84 | 89.23 ± 0.29 | 90.18 ± 0.56 | 90.10 ± 0.35 | 91.34 ± 0.84 | 90.66 ± 0.88 | **93.21 ± 0.19** |
| | Reddit | 70.38 ± 0.28 | 64.66 ± 2.36 | 68.42 ± 0.35 | 73.87 ± 0.72 | 65.85 ± 0.50 | 80.11 ± 0.05 | 78.39 ± 0.45 | **83.20 ± 0.85** |
| | MOOC | 77.34 ± 0.56 | 67.89 ± 2.32 | 89.58 ± 0.58 | 92.31 ± 1.51 | 92.22 ± 0.35 | 91.24 ± 0.34 | 92.65 ± 0.43 | **93.52 ± 0.65** |
| | LastFM | 76.10 ± 0.95 | 72.07 ± 2.29 | 82.42 ± 0.76 | 72.12 ± 3.67 | 81.40 ± 1.04 | **88.09 ± 0.15** | 82.94 ± 0.37 | 87.46 ± 0.39 |
| | Enron | 73.76 ± 1.71 | 63.82 ± 1.55 | 63.82 ± 2.44 | 65.68 ± 2.20 | 75.92 ± 0.55 | 78.76 ± 0.58 | 69.90 ± 1.09 | **86.03 ± 0.64** |
| | Social Evo | 91.30 ± 0.84 | 88.59 ± 0.97 | 98.52 ± 0.13 | 97.97 ± 0.61 | 99.04 ± 0.08 | 98.78 ± 0.04 | 99.14 ± 0.02 | **99.34 ± 0.05** |
| | UCI | 70.87 ± 0.12 | 55.33 ± 4.01 | 79.44 ± 0.55 | 71.71 ± 2.21 | 82.38 ± 3.84 | 83.73 ± 1.32 | 79.59 ± 1.07 | **93.03 ± 0.45** |
| | Avg. Rank | 6.14 | 8.00 | 5.57 | 5.14 | 4.29 | 2.57 | 3.14 | **1.14** |
| ind | Wikipedia | 78.33 ± 0.33 | 73.56 ± 0.84 | 89.23 ± 0.29 | 90.18 ± 0.56 | 90.10 ± 0.36 | 91.34 ± 0.85 | 90.67 ± 0.88 | **93.21 ± 0.19** |
| | Reddit | 70.39 ± 0.28 | 64.66 ± 2.36 | 68.45 ± 0.35 | 73.88 ± 0.72 | 65.84 ± 0.49 | 80.11 ± 0.05 | 78.38 ± 0.45 | **83.20 ± 0.84** |
| | MOOC | 77.34 ± 0.56 | 67.89 ± 2.32 | 89.58 ± 0.58 | 92.30 ± 1.52 | 92.21 ± 0.35 | 91.25 ± 0.34 | 92.66 ± 0.42 | **93.52 ± 0.65** |
| | LastFM | 76.10 ± 0.96 | 72.07 ± 2.29 | 82.42 ± 0.76 | 72.13 ± 3.67 | 81.40 ± 1.04 | **88.09 ± 0.15** | 82.94 ± 0.36 | 87.46 ± 0.39 |
| | Enron | 73.76 ± 1.71 | 63.82 ± 1.55 | 63.81 ± 2.43 | 65.68 ± 2.20 | 75.92 ± 0.55 | 78.76 ± 0.57 | 69.90 ± 1.09 | **86.03 ± 0.64** |
| | Social Evo | 91.30 ± 0.84 | 88.59 ± 0.97 | 98.52 ± 0.13 | 97.97 ± 0.61 | 99.04 ± 0.08 | 98.78 ± 0.04 | 99.14 ± 0.02 | **99.34 ± 0.05** |
| | UCI | 70.89 ± 0.11 | 55.32 ± 4.01 | 79.44 ± 0.54 | 71.75 ± 2.22 | 82.39 ± 3.84 | 83.75 ± 1.32 | 79.60 ± 1.08 | **93.04 ± 0.45** |
| | Avg. Rank | 6.14 | 7.86 | 5.71 | 5.14 | 4.29 | 2.57 | 3.14 | **1.14** |

Table 8: AUC-ROC for inductive link prediction under random, historical, and inductive measurements. The best and second-best results are emphasized by bold and underlined fonts. Note that the results are multiplied by 100 for a better display layout. MS is the abbreviation of measurements.

| NSS | Datasets | JODIE | DyRep | TGAT | TGN | TCL | GraphMixer | DyGFormer | SPACETGN |
|---|---|---|---|---|---|---|---|---|---|
| rnd | Wikipedia | 90.72 ± 0.52 | 87.63 ± 0.56 | 94.40 ± 0.17 | 95.40 ± 0.19 | 94.19 ± 0.28 | 94.88 ± 0.07 | 97.11 ± 0.04 | **97.71 ± 0.13** |
| | Reddit | 92.73 ± 0.41 | 93.43 ± 0.75 | 95.35 ± 0.13 | 95.73 ± 0.24 | 91.77 ± 0.10 | 92.61 ± 0.19 | 96.85 ± 0.10 | **97.07 ± 0.21** |
| | MOOC | 80.86 ± 0.68 | 82.41 ± 0.96 | 85.60 ± 0.31 | **91.68 ± 0.87** | 81.00 ± 0.21 | 80.71 ± 0.17 | 84.89 ± 0.42 | 87.19 ± 0.22 |
| | LastFM | 80.92 ± 1.49 | 82.79 ± 1.47 | 74.49 ± 0.17 | 80.94 ± 1.21 | 74.51 ± 0.83 | 77.33 ± 0.30 | 92.08 ± 0.17 | **92.64 ± 0.13** |
| | Enron | 79.90 ± 1.51 | 76.18 ± 0.84 | 62.24 ± 1.69 | 78.92 ± 2.56 | 81.16 ± 0.81 | 74.89 ± 0.73 | **88.57 ± 0.68** | 87.52 ± 0.57 |
| | Social Evo | 93.10 ± 0.40 | 90.81 ± 0.71 | 92.75 ± 0.10 | 91.78 ± 0.46 | 94.19 ± 0.12 | 93.19 ± 0.09 | 94.96 ± 0.07 | **95.25 ± 0.03** |
| | UCI | 71.93 ± 1.36 | 45.43 ± 5.20 | 74.15 ± 1.80 | 81.07 ± 0.57 | 80.13 ± 6.17 | 85.04 ± 1.79 | 89.85 ± 0.27 | **90.71 ± 0.45** |
| | Avg. Rank | 5.86 | 6.14 | 5.71 | 3.86 | 5.43 | 5.57 | 2.14 | **1.29** |
| hist | Wikipedia | 74.96 ± 0.20 | 71.65 ± 0.98 | 85.67 ± 0.38 | 87.14 ± 0.62 | 86.66 ± 0.57 | 87.82 ± 1.44 | 88.32 ± 1.14 | **91.04 ± 0.32** |
| | Reddit | 65.94 ± 0.57 | 61.80 ± 1.45 | 67.07 ± 0.28 | 69.74 ± 0.68 | 64.23 ± 0.44 | 76.62 ± 0.12 | 74.34 ± 0.44 | **79.35 ± 1.05** |
| | MOOC | 76.14 ± 0.51 | 69.40 ± 2.69 | 88.06 ± 0.75 | 91.58 ± 1.51 | 91.31 ± 0.37 | 90.52 ± 0.41 | 91.78 ± 0.44 | **93.37 ± 0.65** |
| | LastFM | 74.12 ± 0.99 | 69.72 ± 1.59 | 78.14 ± 0.91 | 70.11 ± 2.26 | 79.37 ± 1.72 | **87.28 ± 0.10** | 79.12 ± 0.38 | 83.09 ± 0.65 |
| | Enron | 72.78 ± 2.36 | 63.78 ± 1.39 | 59.98 ± 2.21 | 64.77 ± 1.41 | 72.26 ± 0.80 | 77.63 ± 0.43 | 68.15 ± 1.09 | **82.01 ± 0.62** |
| | Social Evo | 91.24 ± 0.54 | 87.72 ± 0.97 | 98.24 ± 0.14 | 97.64 ± 0.77 | 98.85 ± 0.11 | 98.65 ± 0.06 | 99.00 ± 0.04 | **99.32 ± 0.03** |
| | UCI | 69.33 ± 0.22 | 52.49 ± 3.56 | 73.17 ± 0.93 | 64.65 ± 2.38 | 77.96 ± 4.84 | 79.82 ± 1.49 | 75.28 ± 1.09 | **91.16 ± 0.68** |
| | Avg. Rank | 6.00 | 7.86 | 5.71 | 5.28 | 4.14 | 2.71 | 3.14 | **1.14** |
| ind | Wikipedia | 74.96 ± 0.20 | 71.64 ± 0.98 | 85.68 ± 0.38 | 87.14 ± 0.62 | 86.66 ± 0.58 | 87.82 ± 1.44 | 88.33 ± 1.14 | **91.05 ± 0.32** |
| | Reddit | 65.94 ± 0.57 | 61.80 ± 1.45 | 67.10 ± 0.28 | 69.74 ± 0.68 | 64.22 ± 0.43 | 76.62 ± 0.12 | 74.34 ± 0.44 | **79.35 ± 1.05** |
| | MOOC | 76.14 ± 0.50 | 69.40 ± 2.69 | 88.06 ± 0.75 | 91.58 ± 1.51 | 91.31 ± 0.37 | 90.52 ± 0.41 | 91.78 ± 0.43 | **93.38 ± 0.65** |
| | LastFM | 74.13 ± 0.99 | 69.73 ± 1.59 | 78.15 ± 0.91 | 70.11 ± 2.26 | 79.37 ± 1.72 | **87.28 ± 0.10** | 79.12 ± 0.38 | 83.09 ± 0.65 |
| | Enron | 72.78 ± 2.36 | 63.78 ± 1.39 | 59.98 ± 2.21 | 64.78 ± 1.41 | 72.26 ± 0.79 | 77.63 ± 0.42 | 68.15 ± 1.09 | **82.01 ± 0.62** |
| | Social Evo | 91.24 ± 0.54 | 87.72 ± 0.97 | 98.24 ± 0.14 | 97.64 ± 0.77 | 98.85 ± 0.11 | 98.65 ± 0.06 | 99.00 ± 0.04 | **99.32 ± 0.03** |
| | UCI | 69.30 ± 0.22 | 52.46 ± 3.57 | 73.14 ± 0.94 | 64.73 ± 2.40 | 77.97 ± 4.85 | 79.85 ± 1.48 | 75.30 ± 1.10 | **91.18 ± 0.67** |
| | Avg. Rank | 6.00 | 7.86 | 5.71 | 5.29 | 4.14 | 2.71 | 3.14 | **1.14** |

sults decline without our feature extraction. In conclusion, our Space-Temporal feature extraction distills more accurate temporal information, and the Historical Occurrence feature extraction effectively captures the dynamic graph's cycle information. Together, these tailored feature extraction demonstrate their necessity and effectiveness.

## F  HYPERPARAMETER CONFIG

The parameters used for our comparison model are the optimal parameters in DyGLib Yu et al. (2024), and the parameters used for SPACETGN are shown below.

**SPACETGN:**

- Number of first-hop neighbors $l$ : 20
- Number of time-window series $r$: 64
- Number of historical occurrences $o$: 64
- Dimension of node encoding $d_V$: 172
- Dimension of edge encoding $d_E$: 172
- Dimension of time encoding $d_T$: 100
- Dimension of topological structure encoding $d_{space}$: 50
- Dimension of temporal dynamics encoding $d_{temporal}$: 50
- Dimension of historical occurrence encoding $d_{occur}$: 50
- Dimension of aligned encoding $d$: 50
- Number of MLP-Mixer layers $L$: 2
- Dimension of output representation $d_{output}$: 172

