# OpenReview forum: "SPACETGN: Augmented Mini-Batch Negative Sampling  for Continuous-Time Dynamic Graph Learning"
_ICLR.cc/2025/Conference — ICLR 2025 Conference Withdrawn Submission_

### Official Review · Reviewer_U7Yx · 2024-11-01

**Soundness:** 3
**Presentation:** 2
**Contribution:** 3
**Rating:** 5
**Confidence:** 5

**Summary:**

In this work, the authors investigated how to improve training negative samples used for link prediction in temporal graph learning. To this end, the authors proposed SPACETGN, a CTDG learning framework and an augmented hard negative sampling mini-batch strategy (AMNS). Two new feature extract strategies to extract spatio-temporal subgraphs and historical reoccurrence pattern of nodes are also used. Overall, the added AMNS strategy is interesting and improves the performance of various existing models and SPACETGN achieves good performance too.

**Strengths:**

Overall, I believe the idea to investigate how to select the best negative samples for temporal graph training is very interesting. With the experiments, the authors also showed promising results and improvement when applying AMNS to existing models. Here are the strength of the paper

- An effective negative sampling strategy for training is significant for temporal graph learning as it can be applied to almost all models designed for link prediction. Based on the results provided, AMNS is a promising strategy.

- The proposed AMNS strategy is original and differs from existing strategies seen for evaluation.

- The performance improvement brought by AMNS on existing methods are significant. SPACETGN architecture also achieves good performance

I believe the paper is good but has several limitations I hope the authors would address to improve the paper, please see the weakness and questions section for the discussion. I would be happy to increase the score if some or all of my concerns are addressed.

**Weaknesses:**

Here are the weaknesses of the paper:

- **lack of clarity in writing**, there are some places where it is hard to understand what the authors are trying to express or made wrong statement.
  - a). For example incorrect statement on line 186, "In contrast to setting predetermined random negative samples for each training epoch as described in Huang et al. (2024)". In the TGB work, the authors didn't provide predetermined random negative samples for training but rather just for evaluation. Thus this is incorrect, I would remove this sentence.
  - b). on line 319, "We train the models 100 times over time and use an early stopping strategy with a patience value of 20." This is unclear, did you train 100 models over 100 random seeds, or did you train models with 100 epochs?
  - c). on line 74, what does "information discrimination" mean?
  - d). "space-temporal" should be "spatial-temporal"

- testing only on smallest TGB Dataset. I appreciate that the authors have reported for both MRR and AP / AUC evaluation settings. From the results in Table 3 though, it seems the AP metric is highly saturated with many datasets have close to perfect performance on many methods for example Social Evo. If possible, please include additional results for SPACETGN on TGB datasets such as `tgbl-review`, `tgbl-coin` and the authors can also use the public leaderboard for reference to the performance of existing methods. The focus can be on how much improvement AMNS brings to each model, not necessarily required to be SOTA on the leaderboard.

- **missing SOTA baselines.** There are a few recent methods that are missing. For example, the authors discussed [FreeDyG](https://openreview.net/forum?id=82Mc5ilInM) but didn't include it. There is also the recent work [CTAN](https://proceedings.mlr.press/v235/gravina24a.html). I hope the authors can add the baselines if possible (optional).

**Questions:**

- **Computational Complexity.**  What is the exact computational complexity of the AMNS strategy? It seem to add a significant amount of overhead when compared to random sampling or even historical sampling? The authors should provide a detailed complexity in $O()$ notation.

- the authors claim that random negative leads to overfitting in training however the evidence is not clear to me. Can you further explain this point.

---

### Official Review · Reviewer_SVZf · 2024-11-01

**Soundness:** 2
**Presentation:** 1
**Contribution:** 1
**Rating:** 3
**Confidence:** 3

**Summary:**

This paper focuses on continuous-time dynamic graph (CTDG) learning for link prediction. It proposes an augmented hard negative sampling (AMNS) strategy and a new TGN SPACETGN. The model demonstrates state-of-the-art performance across many datasets within a benchmarking scenario. However, my overall impression of this paper is that it appears hastily written and is difficult to follow. Significant effort would be required to reformulate this paper to reach publishable quality. I believe I have spent enough time (a couple of hours) reviewing this paper, but I cannot understand the intuition behind the designs.

**Strengths:**

It is somewhat novel to address the drawbacks of random, independent, and historical negative sampling through a more refined sampling approach. The experimental results are impressive, demonstrating that the proposed negative sampling strategy and TGN framework can improve upon or outperform existing state-of-the-art methods.

**Weaknesses:**

W1. The mathematical notations in the manuscript are difficult to follow. The current version transitions between notations without sufficient mathematical formulation or clarity. For instance, in lines 206-207, it is unclear what constitutes a positive candidate, a historical candidate, and a random candidate. The explanation cannot rely on the ambiguous Figure 2, which provides little informative value. I also find it hard to understand what lambda is in equation 1 and what this rating function is used for.

W2. The flow of the proposed sampling method and the TGNN is unclear, and the titles of Figures 2 and 3 lack informativeness. To enhance clarity and ensure readers understand the roles and interactions of each component, these should be explained in the main manuscript. Currently, much of the detailed information is found in Appendix A (just with "Detailed descriptions of SPACETGN are
available in Section A." from the main page, line 293), which is optional for reviewers and future readers. You cannot expect readers to rely on the appendix for essential details. The main pages should be self-sufficient and provide comprehensive explanations for SpaceTGN. Specifically, the bullet points in lines 262-285 are regarding (b) Extracting Key Sequences in Figure 3, but I cannot find any explanations on (c) Encoding Features and (d) Mixing Features in the main pages.

W3. One advantage of traditional negative sampling is its efficiency. However, the new negative sampling design requires extensive computations each time a new sample is generated. The paper lacks theoretical analysis or experimental comparisons regarding the impact on training time, which should be addressed to understand the trade-offs in computational efficiency.

W4. Code seems not available; only a few hyperparameter configs are mentioned in Appendix F for reproduction.

W5. The paper requires thorough proofreading. I recommend that the authors pay close attention to grammatical and syntactic errors, as well as typos. The current version feels somewhat rushed, with notable issues in the flow and semantics that should be addressed to improve readability and coherence. Some minors I identified:

Minors:
- typo: line 018 "with a augmented"
- Line 152. The Continuous-Time Link Prediction task itself has nothing to do with representations h_u and h_v. There are many works even before graph representation learning that deal with temporal link prediction.
- There seems to be a layout problem under Figure 2.
- The sentence in lines 211-213 seems incomplete?
- line 228 Figure 3: Framework of the proposed what?
- line 292 Missing space: "timestamp.Detailed"

The list of minor errors provided above is likely incomplete, and there are probably many more that need to be fixed.

**Questions:**

Please refer to weaknesses.

Although I am familiar with the topic, I found it challenging to follow the presentation of this work, and I am not confident that I fully understand the proposed methods. Consequently, I am assigning a confidence rating of 3.

---

### Official Review · Reviewer_iQA8 · 2024-11-03

**Soundness:** 4
**Presentation:** 1
**Contribution:** 2
**Rating:** 3
**Confidence:** 4

**Summary:**

This paper proposes an augmented mini-batch negative sampling strategy for dynamic graphs by sampling hard negative samples and mitigate the challenges posed by false positives and false negatives. This paper also presents a novel time-sequence-based dynamic graph learning model utilizing the space-temporal dependency and historical occurrence.

**Strengths:**

1. This paper proposes a novel negative sampling strategy for dynamic graph.
2. The experimental results show the great improvement compared to the random negative sampling strategy.

**Weaknesses:**

1. The presentation of this paper needs great improvement due to a large number of typos.
- There is an extra whitespace in Line 200.
- From line 211 to line 213, the timestamp  (i.e., t11) in this example is different from the timestamps (i.e., t0-t4) in figure 2, resulting in confusion. Do you mean figure 3?
- In the caption of figure 3, it seems that the name of the proposed method is missing in "framework of the proposed ".
- In equation 1, it seems that $r_{u,v}$ should be $r_{u,v}(t)$.
- The experimental results in Table 3 are hard to read as the whitespace between any two columns is too small.
- Please use \citep instead of \cite in the latex. A whitespace appears in some \cite before the cited content, but it doesn't in some places, such as Line 327, Line 328, Line 792, Line 795, line 798, etc.
- Please move the figure 5 to somewhere else. Do not leave one line of text (i.e., line 387) between figure 4 and figure 5, which can be easily considered as the caption in figure 4 and then ignored by the readers. Additionally,  it seems that figure 6 in "As shown in figure 6" in line 387 should be figure 5.

2. In section 4.2, the title of it is related to CTDG architecture, while most of the content is related to the historical sequences sampling. The authors move the architecture to appendix A. The authors should reorganize the paper.

3. The authors should explicitly mention what the x-axis in figure 5. Without this information, it's hard to understand the discussion in section 5.4, which could be one of the major experimental results validating the adaptivity of the proposed method.

**Questions:**

Q1: In figure 3, you consider the pair of nodes u and c as a hard negative pair at timestamp $t_{11}$. What if the pair of nodes u and c can form a positive pair in timestamp t_{12} as they were connected before and they are very likely to be connected again in a later timestamp?

Q2: What is the exact function for $r_{u,v}(t)$? How does it measure the similarity of node pair u and v. What kind of feature does it used for measurement?

Q3: What is the x-axis in figure 5?

---

### Official Review · Reviewer_iSH5 · 2024-11-03

**Soundness:** 1
**Presentation:** 1
**Contribution:** 3
**Rating:** 3
**Confidence:** 4

**Summary:**

The authors formulate a negative sampling strategy intended to improve continuous-time temporal link prediction performance by providing more difficult negative links than current methods. The idea is to make negative samples more similar to the positive edges, which should lead to a harder prediction problem, requiring temporal link prediction methods to learn better representations that facilitate more accurate predictions. Guided by three criteria that hard negative links should satisfy, the authors define a rating function to guide what they call "augmented mini-batch negative sampling". To test their approach, the authors developed SpaceTGN, a "space-temporal locality aware continuous time dynamic graph architecture" based on MLPs, and tested it on seven real-world networks against eight baselines. In their test, the baselines are also provided with the enhanced (harder) negative samples during training. The results show that their approach outperforms the baselines in most cases. Moreover, an analysis of the negative links suggests that, compared to conventional negative sampling, the authors's approach generates negative links that are more similar to the positive edges.

**Strengths:**

- The authors consider an important research question and address it in a way that makes sense, that is, by generating harder training instances.
- The evaluation on empirical data demonstrates the effectiveness of the proposed approach.
- The proposed negative sampling approach was also used in combination with the baselines.

**Weaknesses:**

- Several parts of the text remain unclear, even after reading them many times. For example, I am still unsure what the authors are trying to say when they describe the criteria that hard negative samples should satisfy. I would recommend revising the criteria for clarity.
- The authors claim that SpaceTGN "achieves significant performance improvement [...] than other baselines", however, they do not provide a statistical analysis to support the claim of significance.
- The paper does not fully deliver on what the abstract appears to promise. Specifically, the abstract mentions that "SpaceTGN achieves state-of-the-art performance in most datasets, demonstrating its effectiveness in ameliorating model bias", however, the paper neither discusses model biases nor how SpaceTGN ameliorates them.

**Questions:**

1. In l. 69 it is mentioned that "[...] it is necessary [...] to balance the divergence in negative sample distribution in the training and testing stages". Since neither "balancing" nor "divergence" are mentioned later on in the paper I find it hard to follow what it is that should be balanced. Is it the balance between hard and easy negative edges in the training and test sets? I would find that surprising since it was my understanding that the whole point is to replace the easy negative samples with harder ones.
2. What is meant by the "statistical performance" of models (l. 102)?
3. If I understand the explanations correctly, one aspect of hard negative edges is that they should be quite similar to true positive edges. However, since real-world data is often incomplete, it may be possible that an edge has simply not been observed even though it existed at some point in the past, that is, it is a false negative. So how do we know that the hard negative samples are not simpy false negatives? Is the assumption that the training data is complete?
4. What are "soft negative samples" (mentioned in l.202)?
5. What is the short example mentioned in l. 211-213 intending to show? There are interactions at time $t_{11}$ in Fig. 2 mentioned, however, Fig. 2 does not show that timepoint. Moreover, the sentence appears to be incomplete.
6. The rating function shown in Eq. 1 involves the learned node embeddings, meaning that estimating a link's difficulty depends directly on learning meaningful node embeddings. Could this mean that there is a cold start problem or a minimum amount of observations required before it is possible to generate hard negative samples?
7. Lines 267 and 284 mention that excess neighbours and too-long sequences are truncated. How are they truncated? For example, are the most/least recent interactions ignored?
8. There are a couple of sentences in section 5.1 that are unclear, namely:
  - "We optimized all Adam models" (l. 318) -- What are Adam models? Or is this supposed to say that all models were trained using the Adam optimiser?
  - "We train the models 100 times over time" (l. 319) -- Does this mean that all models are trained a hundred times using the same data? If yes, are the same negative samples used for each of the 100 iterations?
  - "We run these methods with 0 to 4 seeds five times and report the average performance to eliminate bias." (l. 322-323) -- Does this mean that the seeds {0,1,2,3,4} are used for five runs? How does this facilitate reducing model bias?
  - The second paragraph in section 5.5 discusses the results shown in Table 3, which shows the average precision of the tested methods. However, the text mentions accuracy. Which one is it, average precision or accuracy?
9. Fig. 6 shows the cosine similarity between positive and negative samples. What are the cosine similarities based on? Are they based on the embeddings that SpaceTGN has learned for them?

Minor points
- The caption of Fig. 3 seems to be missing a word, perhaps "approach" or "method"?
- The y-axes in Fig. 5 have different scales, which makes the figure somewhat harder to interpret. I recommend using the same scale in all subplots. Moreover, is it correct that the density can take on values higher than 1?
- The results in Table 3 are hard to read because the space between the columns is so small.

---

### Note · Authors · 2025-01-16

**Comment:**

I have read and agreed to the venue's withdrawal policy on behalf of myself and my co-authors. We sincerely thank the anonymous reviewers for their insightful comments. The suggestions will be reflected in our future version.

**Withdrawal Confirmation:**

I have read and agree with the venue's withdrawal policy on behalf of myself and my co-authors.